# Olive Mill Wastewater Remediation: From Conventional Approaches to Photocatalytic Processes by Easily Recoverable Materials

**Melissa G. Galloni [1,2], Elena Ferrara [1], Ermelinda Falletta [1,2,*] and Claudia L. Bianchi [1,2]**

**Abstract:** Olive oil production in Mediterranean countries represents a crucial market, especially for Spain, Italy, and Greece. However, although this sector plays a significant role in the European economy, it also leads to dramatic environmental consequences. Waste generated from olive oil production processes can be divided into solid waste and olive mill wastewaters (OMWW). These latter are characterized by high levels of organic compounds (i.e., polyphenols) that have been efficiently removed because of their hazardous environmental effects. Over the years, in this regard, several strategies have been primarily investigated, but all of them are characterized by advantages and weaknesses, which need to be overcome. Moreover, in recent years, each country has developed national legislation to regulate this type of waste, in line with the EU legislation. In this scenario, the present review provides an insight into the different methods used for treating olive mill wastewaters paying particular attention to the recent advances related to the development of more efficient photocatalytic approaches. In this regard, the most advanced photocatalysts should also be easily recoverable and considered valid alternatives to the currently used conventional systems. In this context, the optimization of innovative systems is today's object of hard work by the research community due to the profound potential they can offer in real applications. This review provides an overview of OMWW treatment methods, highlighting advantages and disadvantages and discussing the still unresolved critical issues.

**Keywords:** olive oil production; olive mill; wastewater remediation; polyphenols; conventional photocatalysts; magnetic photocatalysts; floating devices; environmental remediation

# 1. Introduction

Olive oil production is a fundamental sector for several European (EU) States, especially Spain, Italy, and Greece. In particular, Spain has the largest area of olive cultivation (estimated at *ca.* 2.47 million ha), followed by Italy (*ca.* 1.16) and Greece (about 0.81 million ha) [1,2]. However, olive oil production is responsible for several environmental concerns (soil contamination, underground seepage, water-body pollution, and odor emissions) due to poor waste management practices [3]. In this scenario, concerning olive mill wastewaters (OMWW), special attention must be paid to their high phenolic content, which is responsible for their antibacterial effect, phytotoxic effect, and dark colour.

Recently, phenols, fatty acids, and volatile acids have been recognized as potentially hazardous for environmental health: the former have pronounced antimicrobial and phytotoxic properties, whereas the latter show toxicity due to their long alkyl chain.

All these components make OMWW toxic to anaerobic bacteria, thus inhibiting conventional secondary and anaerobic treatments in municipal water plants. Furthermore, the high BOD (biological oxygen demand) and COD (chemical oxygen demand) levels, which

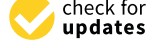



cannot be reduced by anaerobic digestion, represent a further threat to receivers [2,4]. Moreover, land spreading and treatment in evaporation ponds could lead to problems related to groundwater pollution. The use of olive oil waste in agriculture may also affect the acidity, salinity, N immobilization, microbial response, leaching of nutrients, and concentration of lipids, organic acids, and phenolic compounds [5].

Alternative approaches based on physical treatments, such as dilution, evaporation, centrifugation, or sedimentation guarantee a high level of OMWW purification. However, they are expensive and energy-consuming, thus leading to an exponential increase in the processing cost. The olive oil industry, in its current status, composed of small and dispersed factories, cannot bear such high costs [6–13].

In recent years, advanced oxidation processes (AOPs), including photolysis, photo-oxidation, Fenton, and photo-Fenton reaction, have emerged as promising alternatives for simplicity and high organic removal efficiencies [14–20]. In particular, heterogeneous photocatalysis seems to be a successful technology in water decontamination due to its non-toxicity, low cost, and mineralization efficacy. However, due to the OMWW matrices' complexity, it is not easy to develop and successively optimize efficient photocatalytic systems that are so far characterized by common limitations (i.e., difficult recovery, poor stability, low reusability, fast deactivation).

Based on these premises, in the present work, for the first time, we illustrate the conventional methods commonly used to treat OMWW along with their related advantages and limitations. Then, a critical insight on alternative strategies for developing efficient photocatalytic systems based on recoverable catalysts is proposed. The latters can be used as alternatives to conventional photocatalysts. This topic is of fundamental importance for the research community as shown by the hard work currently been done for developing novel devices with high potential in real applications, acting as a bridge between environmental protection and circular economy.

## 2. An Insight into the EU Legislation

Olive oil is the desired product of the olives industry. Unfortunately, olive mill pomace and wastewater represent undesired by-products, requiring proper disposal treatments because of their complex composition (Figure 1).

The present work aims at discussing only the production and treatment of OMWW. OMWW composition is influenced by different factors, i.e., extraction methods, olives' type and origin, climate conditions, and cultivation/processing practices [21]. In general, it can be mainly summarized as follows (Figure 1): *ca.* 80–83 wt.% consists of water, *ca.* 15–18 wt.% relates to organic compounds (mainly polyphenols, phenols, and tannins), and the remaining 2 wt.% contains inorganic matter (i.e., potassium salts and phosphates). Specifically, phenols levels in OMWW range from 1 to 8 $g \cdot L^{-1}$, whereas micronutrients and mineral nutrients mainly consist of $K_2O$, and $P_2O_5$, which can be found in considerable amounts (2.4–10.8 or 0.3–1.5 $g \cdot L^{-1}$ intervals, respectively) [2]. Thus, it is critical to design efficient treatment methods, aligned to precise legislative constraints, whose general panorama is described below.

Concerning the processing of olive residues, the reform of standard agricultural policy related to olive oil does not provide specific provisions for their management [1]. It should be noted that a significant part of EU legislation acts according to Directives. These latter are legislative acts, setting objectives that all EU countries must reach and translate into their national legislation. This means that the member Countries have to adopt and impose complementary measures that should be compliant with the EU directives.

Following this scenario, an example is setting the emission limits and environmental quality standards. Of course, every Country can adopt laws and regulations that can be very different compared to others. Still, in the end, international norms are necessary for a common strategy to manage olive waste. In general, EU legislation governs each member state's framework of national legislation. Several EU laws regulate waste management, and the Waste Framework Directive, WFD (2008/98/EC), acts as core legislation, including

hazardous waste and oil rules [22]. In addition, Landfill Directive 99/31/EC regulates landfill disposal [23]. In this case, the waste producer, such as the olive mill operator, is responsible for managing wastes up to their recovery and disposal [24].

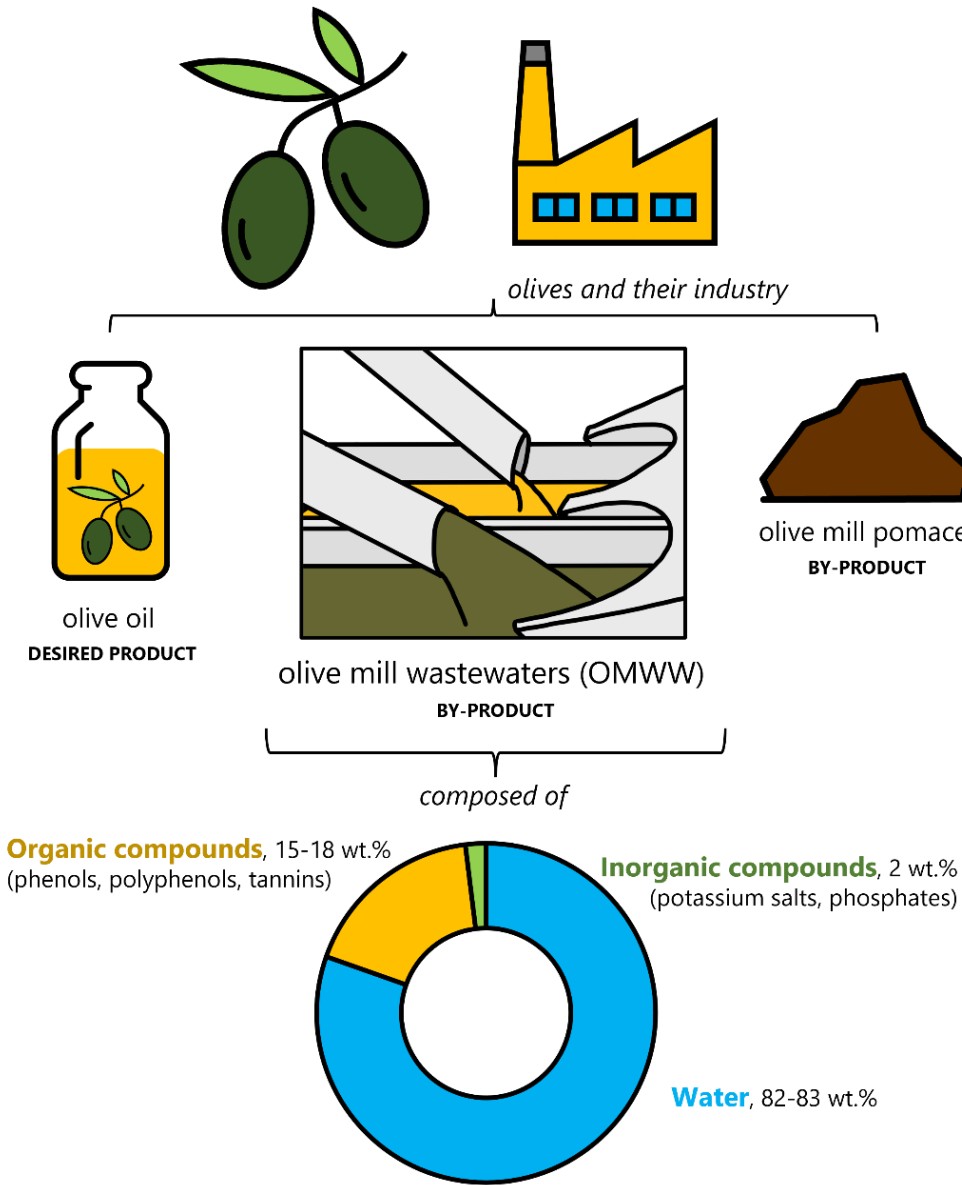

**Figure 1.** Scheme of products and by-products from the olive oil industry.

Here, the crucial point is to classify wastewaters as waste or by-products. If they are considered by-products, their further use as fertilizers with few restrictions is strongly recommended [25]. In this context, the EC Directive 2008/98 (point 22) clarifies the necessity to discriminate well between "waste" and "by-product", but unfortunately, considerable confusion is still present [22]. So, in many cases, law courts have to solve specific issues. To summarize, no EU legislation related to the management of OMWW exists today, and each EU country sets precise standard parameters.

## 3. Emerging Innovative Approaches for Olive Oil Production

Conventional techniques in olive oil extraction have not significantly changed in the last 25 years. Three main steps can be identified (Figure 2): crushing and malaxation, which mainly affect the oil quality and yield, and centrifugation [26,27].

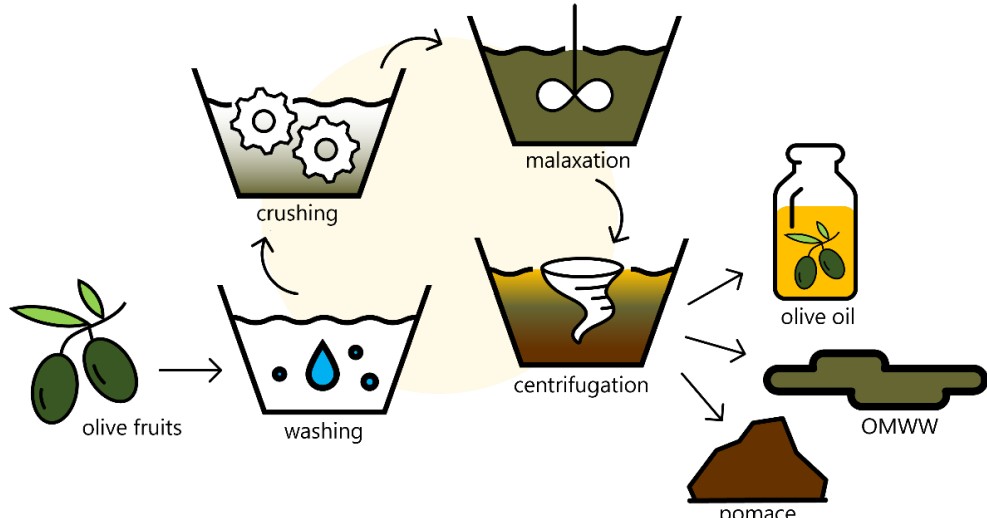

**Figure 2.** Scheme of olive oil production.

At first, stems, twigs, and leaves are separated from olive fruits [28]. These latter are then washed in a proper plant to remove dust, dirt, etc. In some plants, the washing water is recycled for the process after solid sedimentation or filtration, whereas in other cases, olives are directly processed without the washing step [29]. The next step involves malaxation: olives are ground up, mixed with/without their stones, and put in tanks, where the paste is divided into vegetation waters, pomace, and oil. Pomace, a brown-colored residue, is obtained by centrifugation and sedimentation after pressing olives [30,31]. Pomace mainly consists of skin pulp and pit fragments. Its separation is carried out using a horizontal decanter centrifuge and an olive oil press. The centrifuge step can be performed in two- or three-phases (Figure 3).

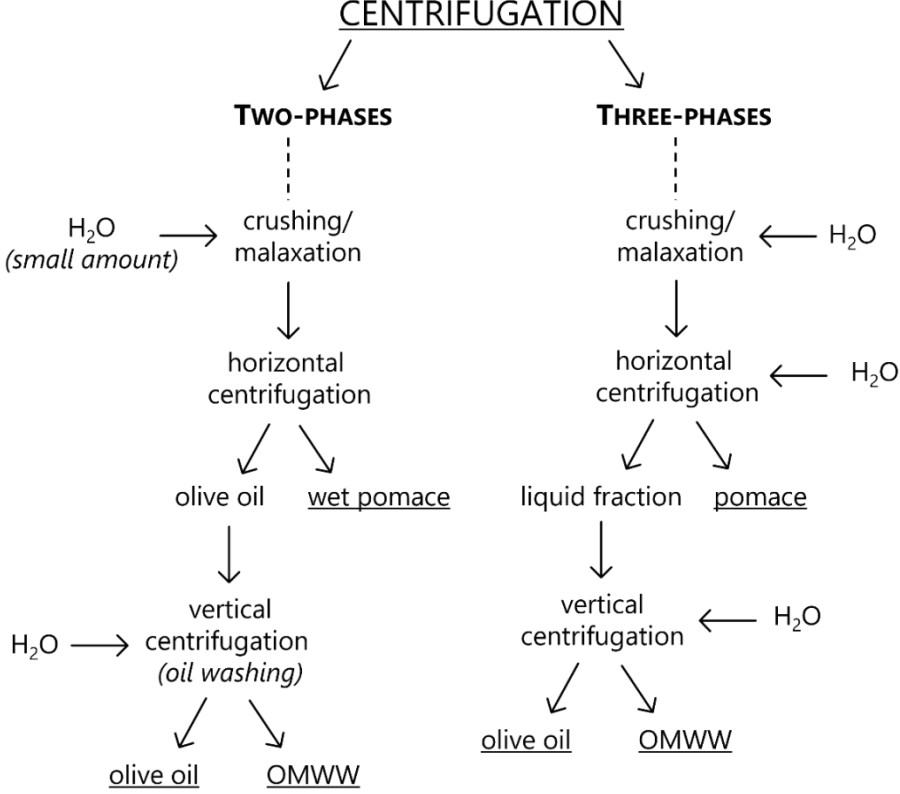

**Figure 3.** Scheme of two-phases and three-phases centrifugation strategies.

In the former case, wet pomace (also known as two-phase olive mill waste, TPOMW) and olive oil are obtained by horizontal centrifugation. Then, the obtained oil is centrifuged with water, producing olive oil and a small stream of OMWW [32,33]. In the latter, the olive paste is divided into pomace and a liquid fraction (olive oil *plus* OMWW), which is centrifuged with water to obtain high-quality olive oil and OMWW [32,33].

However, given the ever more urgent market demand, interesting novel methods characterized by minimal processing are currently the object of study. These approaches aim to obtain a final product with the same nutritional qualities in less time. In this context, numerous solutions, including the use of microwave, high-pressure processing, pulsed light, radio frequency, Ohmic heating, ultrasound, and pulsed electric field (PEF), have been investigated thanks to their advantages (enhanced extraction efficiency in reduced time with increased yield, and low energy consumption) [26,34–37].

Among them, ultrasound emerged as a powerful technology widely used in several extraction processes [37,38] and food processing methods (i.e., emulsification, filtration, crystallization, enzymes' and microorganisms' inactivation, thawing) [39,40]. Ultrasound can be applied to the olive paste to induce oil release from vacuoles in lower malaxation time. It has been demonstrated that high oil quality and yield are obtained [36,41–45].

Pulsed electric field (PEF) technology, used mainly in food science since 1960, consists of exposing food products (solid or liquid) to an electric field, inducing pore formation in cell membranes [46]. Recently, it has demonstrated its efficiency in reversible or irreversible permeabilization of cell membranes in different plants without causing significant temperature increase [34]. The possibility of maintaining low operating temperatures during the oil extraction process represents a valuable goal, as it allows the preservation of the product's organoleptic and nutritional characteristics.

An alternative to the two previous processes is microwave-assisted extraction (MAE), which represents a more efficient and successful strategy than the conventional ones because microwaves provide rapid heating and biological cell structure destruction. As a result, it leads to high-quality products with shallow energy requirements, inducing reduced environmental impact and financial costs [47].

Recently, emphasis has been placed on obtaining an increased Extra Virgin Olive Oil (EVOO) quality, preserving its sensory characteristic and favorable health properties. The quality of the EVOO strongly depends on the presence of phenolic and volatile compounds [43,44]. So, the development of emerging technologies to increase the oil yield while protecting and improving the bioactive oil compounds and quality is of fundamental importance.

Table 1 summarizes some interesting studies related to innovative technologies applied to olive oil extraction, including the maximum extraction yield obtained (i.e., the percentage value given by the ratio between the weights of the extracted oil and olives).

**Table 1.** Emerging extraction methods for olive oil production. Adapted from Ref. [48].

| Olives' Variety | Used Technology [a] | Investigated Parameters | Dependent Variables | Maximum Extraction Yield (%) [b] | Ref. |
|---|---|---|---|---|---|
| Edremit | HPU | Ultrasound time, ultrasound temperature, malaxation time | Oil yield, acidity, peroxide value, and antioxidant properties | 9 | [26] |
| Coratina | HPU | Ultrasound application step (After crushing/before crushing) | Olive paste temperature, energy balance, oil yield, quality indices of oil, minor compounds | 16 | [36] |
| Picual | HPU | Direct/indirect application of ultrasound | Olive paste temperature | 16 | [49] |

**Table 1.** *Cont.*

| Olives' Variety | Used Technology [a] | Investigated Parameters | Dependent Variables | Maximum Extraction Yield (%) [b] | Ref. |
|---|---|---|---|---|---|
| Picual | HPU | Continuous ultrasound application before centrifugation | Oil yield, quality indices, volatile and minor compounds, fatty acid composition | 53 | [50] |
| Edremit, Gemlik, Uslu | HPU | Ultrasound and malaxation time | Oil yield, UV absorbance values acidity, peroxide value, total phenolic content | 68 | [51] |
| Picual | HPU | Olive paste flow, HPU intensity, fruit temperature, olive moisture, and fat content | Olive paste temperature | 17 | [52] |
| Ogliarola Barese | HPU, MW | Thermal effect of US and MW | Malaxation time, oil yield, quality characteristics, and energy efficiency | 17 | [45] |
| Arbequina | PEF | PEF application | Oil yield, acidity, quality characteristics, total phenols, sensory properties | n.d. [c] | [35] |
| Chemlal | MW | Extraction time, acetic acid content in hexane, irradiation power | Oil yield, total phenols, quality parameters | 6 | [53] |
| Peranzana | MW | Malaxation time and MW | Energy consumption, oil yield, structure modifications of olive pastes | n.d. [c] | [54] |
| Coratina | HPU | Sonication time | Oil yield, oil quality indices, phenolic composition | n.d. [c] | [55] |

[a] HPU: high-power ultrasound; PEF: pulsed electric field; MW: microwave; [b] expressed as percentage given by the ratio between the weights of the extracted oil and the olives; [c] not defined.

## 4. Olive Mill Wastewater Treatment

As reported above, OMWW is the waste of olive oil production characterized by high organic content and phytotoxic features mainly due to the presence of phenols, which are responsible for the olive oil's antimicrobial and antioxidant qualities. This makes waste biodegradation difficult in conventional treatment facilities (e.g., anaerobic digestion processes) that generally use microorganisms for waste biodegradation.

According to these premises, the OMWW treatment has faced several traditional approaches, which can be categorized as: disposal, physicochemical, biological, and advanced oxidation methods. Figure 4 schematizes their potentialities and weakness.

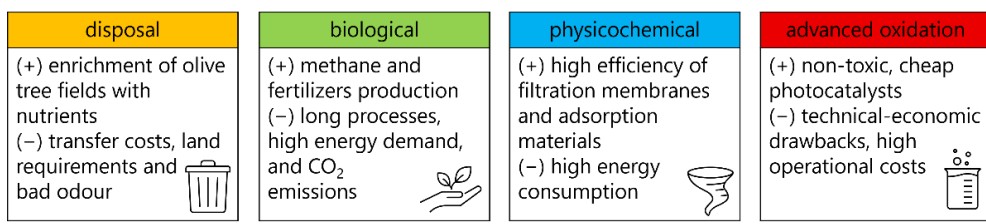

**Figure 4.** Main advantages/disadvantages of the standard technologies used for OMWW treatment.

As can be observed, all the mentioned technologies have specific advantages, but also cost problems. Therefore, some alternatives are currently the object of hard work by the research community. Among all the possibilities, one emerging and valid strategy is the OMWW steam reforming (OMWWSR), which permits valorization of wastes, producing green $H_2$, following the circular economy perspective [32]. It is a promising strategy in the view of the future projections of the $H_2$ market demand recently published by the International Energy Agency [56–58]. This method is described as having a high potential because of the environmental attractiveness of $H_2$, which is able to reduce $CO_2$ emissions in the atmosphere [56]. In this way, OMWWSR could contribute to air pollution reduction and, at the same time, valorize the waste from the olive oil industry [32]. However, this process still deserves to be properly studied and optimized because of some drawbacks affecting the catalyst formulation (e.g., low stability, deactivation, coke poisoning) [32].

The following paragraphs provide an accurate description of each traditional technology.

### 4.1. Disposal Methods

They mainly consist of treatment with calcium oxide (neutralization and coagulation) followed by the disposal of waterproof lagoons [59]. Unfortunately, they are affected by some disadvantages, such as foul odors, mosquitoes, and transfer costs because they require land far from residential areas. As an alternative, OMWW can be carried to fields of olive trees and then spread, thus enriching the soil with nutritive compounds [60].

### 4.2. Biological Methods

Bioremediation is a technology that exploits the metabolic potential of microorganisms to purify contaminated sites. It can be performed in a non-sterile and open area containing numerous organisms. Among these, bacteria have a central role in the process thanks to their ability to degrade pollutants. In addition, fungi and other components (e.g., grazing protozoa) can also affect the process [61]. All these species require nutrients (i.e., carbon, nitrogen, phosphates, metal traces) to survive, so they break down organic compounds to attain them. Bioremediation can occur under aerobic or anaerobic conditions [62]. In the former case, the survival of microorganisms is mainly due to the consumption of atmospheric oxygen. In contrast, in the latter, microorganisms gain food by breaking down chemical compounds in the soil [27].

In recent decades, OMWW has been used in this way, acting as substrate for microorganisms' growth by providing nutritive substances. Some yeast species (i.e., *Candida tropicalis, Yarrowia lipolytica*) together with bacteria of the *Azotobacter vinelandii, Pseudomonas, Sphingomonas, Ralstonia* species have been proven to be helpful in the OMWW aerobic biodegradation and detoxification [63–65]. By way of example, the activity of a free-living $N_2$-fixing bacterium, *Azotobacter vinelandii*, was investigated. In particular, OMWW was initially treated with calcium hydroxide to achieve the pH value of *ca.* 8–10 (stage I). Successively, it was mixed in a bioreactor in the presence of *Azotobacter vinelandii* (stage II). The process carried out according to the procedure reported by Arvanitoyannis et al. [27] resulted in an increasing level of nitrogen and its ammonium form throughout the whole remediation period. On the other hand, regarding phenols and sugar degradation, ca. 66–99% and 100% of phenols'abatement was observed after 3 and 7 days, respectively, whereas sugars were wholly degraded in only 3 days. Low phytotoxic features characterize the final product so it can be exploited as fertilizer [66].

In general, to reduce the high content of phenolic compounds in OMWW, water dilution represents a suitable strategy for a successful aerobic treatment. In fact, phenols are responsible for inhibition of microorganism growth [67,68]. Alternatively, OMWW could be mixed with additional waste and digested with the help of a solid substrate (i.e., straw, sesame bark, olive leaves, vineyard leaves, wood chips, animal manure) [69–71]. Then, when the phenol content of waste decreases, usually after 6–7 months, the final product can be exploited as fertilizer, giving profit [70,71]. Additionally, the composting stage could be coupled with physicochemical processes [72–74]. This last method requires high energy

demand and consequent high $CO_2$ emissions. However, the energy demand can be reduced thanks to simultaneous methane production [75].

### 4.3. Physicochemical Treatments

Among physicochemical treatments, dilution, evaporation, sedimentation, filtration, and centrifugation are commonly used to treat OMWW.

OMWW dilution is usually employed before biological treatments with the final aim to reduce its toxicity to microorganisms. On the other hand, evaporation and sedimentation result in a concentrated OMWW (*ca.* 70–75% more concentrated) thanks to both phase separation/dehydration and organic matter degradation [6,7]. In this context, solar distillation applied to OMWW can remove 80% COD in the distillate in 9 days, maintaining 25% water content [8].

Other strategies have also been investigated, mainly consisting of irreversible thermal treatments. This is the case with combustion and pyrolysis that require a reduced volume of waste and provide energy recovery. Still, unfortunately, they need expensive facilities, emit toxic substances into the atmosphere, and require an OMWW pre-concentration step [9,10].

Centrifugation and filtration increase the effluent pH and conductivity, removing the organic matter using phase separation and exclusion. Ordinarily, combining physical processes, coupled with coagulation/flocculation or adsorption techniques, gives rise to more efficient removal of organic matter. For example, it was found that when the sedimentation is followed by centrifugation and filtration, 21% and 15% decrease in COD and BOD, respectively, was observed, with the further 16% reduction in BOD due to the final filtration [11]. OMWW adsorption on activated clay causes an additional 71% COD reduction. However, a particular focus has to be put on the adsorption/desorption equilibrium since organic and phenolic features start to desorb after a precise contact time. The combination of treatment stages, i.e., settling, centrifugation, filtration, and adsorption on activated carbon, induce a maximum of 94% phenol abatement and 83% organic matter removal [12].

Regarding filtration, it is fundamental to point out that, besides the high efficiency of membranes, these processes require high operative pressures and energy consumption. However, proper membranes can be exploited to recover valuable by-products, such as phenols, which are mainly required for the pharmaceutical and chemical industry [13].

Lime treatment has been selected as a pre-treatment step for reducing OMWW polluting effect due to its inexpensiveness [76–79].

In this context, coagulation-flocculation is a very similar technology to lime treatment. Different coagulants (i.e., ferric chloride, polyelectrolytes) can be exploited [80]. On the other hand, electro-coagulation mainly consists of the suspension and precipitation of charged particles in the waste thanks to an applied voltage. Since this process is characterized by low cost and energy consumption, it is not so efficient in removing organic waste species.

### 4.4. Advanced Oxidation Methods

The addition of strong oxidizing agents can influence the efficiency of wastewater treatment, mainly in terms of the breakdown of recalcitrant and toxic compounds. In this context, high mineralization levels can occur depending on the oxidizing power of the agent employed and contact time. In recent decades, the scientific community has addressed the efforts to exploit advanced oxidation processes (AOPs) to treat industrial effluents and OMWW [14,15]. In general, AOPs combine ozone ($O_3$), light irradiation (UV, solar, visible), hydrogen peroxide ($H_2O_2$), and/or catalysts to produce unstable radical species able to degrade both organic and inorganic pollutants.

In electrolysis, the oxidation of the content of organic species directly occurs on the anode or indirectly by oxidizing agents present in the solution [81]. Over the years, several materials for anodes' production have been studied (i.e., Pt/Ir, Ti/IrO$_2$, Pt/Ti, and boron-doped diamond) [81–84]. However, this is a high-energy consuming approach. In contrast,

Fenton oxidation is based on the addition of Fenton's reagent ($H_2O_2$ and Fe(II)) into the waste [16]. In this case, the oxidation process is due to a cascade of different reactions in the solution. Although it is low energy consumption, $H_2O_2$ makes this technology quite expensive. The photo-Fenton method is very similar to the Fenton one, but the UV radiation accelerates $Fe^{2+}$ regeneration, enhancing, as a consequence, the process efficiency. However, the necessity to employ UV radiation causes high energy consumption [85]. Supercritical water oxidation consists of waste oxidation in the presence or absence of catalyst above the water critical temperature and at high pressures [86–88]. It is a very efficient technology for organic content reduction, but the energy consumption is high due to the high temperatures and pressures required. Finally, ozonation employs $O_3$ as oxidant species for waste oxidation. It is not so efficient in the organic content reduction, but that of phenols is high. Unfortunately, using $O_3$ increases the process costs [89–91].

### 4.4.1. Photocatalytic Treatments

Photocatalysis can be described as an advanced oxidation process able to fully mineralize the contamination in liquid as well as the gas phase under room pressure and temperature [92]. Its efficiency is mainly due to the capability to generate powerful oxidizing agents [14,15,93,94]. In this way, the chemical transformation rate is enhanced by the chosen photocatalyst under light irradiation [95]. Following these perspectives, photocatalysis has found a successful application in the water decontamination field [96], providing promising results in the removal of a large variety of contaminants (e.g., aromatics, pesticides, drugs, oils) [97].

In this context, photocatalytic treatments can be applied in the field of OMWW degradation using both homogeneous and heterogeneous photocatalysts in the presence of UV, visible and solar light irradiation. In this class of treatments, photo-Fenton and solar-Fenton processes are also included [18,98,99].

In the following paragraphs, deeper insights into the current approaches used in the literature are reported with the aim of fully describing the scenarios related to these technologies.

### UV Photocatalysis

Data summarized in Table 2 show how UV photocatalysis finds application in the OMWW treatment in the presence of both homogeneous and heterogeneous catalysts.

**Table 2.** State of the art of UV photocatalysis used to treat OMWW. Adapted from Reference [100].

| OMWW Origin | Type of Process Treatment and Scale | Obtained Results | Ref. |
|---|---|---|---|
| Jordan | (i) $O_3$/UV or (ii) UV/$O_3$, followed by (iii) biodegradation—laboratory scale | COD removal efficiencies up to (i) 91% by UV/$O_3$ followed by biodegradation | [101] |
| Greece | Photocatalytic treatment with $TiO_2$ (Degussa P25)—laboratory scale | 200 mg·$L^{-1}$ COD residual and complete total phenol removal | [98] |
| Spain | pH-temperature flocculation + ferromagnetic core $TiO_2$ + UV photocatalysis— laboratory and pilot scale | 58.3% COD and 27.5% total phenols removal efficiencies; overall COD removal efficiency up to 91% | [100,102] |
| Portugal | nano-$TiO_2$ immobilized in nonwoven paper— laboratory scale | $90.8 \pm 2.7\%$ removal of the phenolic content | [103] |
| Italy | UV/$TiO_2$— laboratory and pilot scale | COD reduction around 50% upon 1.5 g·$L^{-1}$ nanocatalyst dosage | [104,105] |

Since 1972, titanium dioxide ($TiO_2$)-based photocatalysts have been investigated [106] and then widely used for their effective semiconductor features, enabling the removal of various pollutants in environmental remediation [107–109]. Interesting properties characterize these systems, like chemical stability, long-term stability, remarkable oxidation ability, and low-cost [110–112]. Heterojunction photocatalysts based on $TiO_2$ have been studied mainly for the mineralization of targeted pollutants into harmless products, thanks to the

generation of electron-hole ($e^-$/$h^+$) pairs if the semiconductor is under UV radiation [97]. In this frame, 2.80 V oxidizing power was produced by hydroxyl radicals produced during the photocatalytic step [96]. Besides the high chemical and physical stability of $TiO_2$, this material tends to go through phase transformation from anatase to rutile [113]. This induces a detrimental effect on the resulting $TiO_2$-materials because the rutile-phase has a lower surface area, negatively impacting the photocatalytic behaviour because of the ($e^-$/$h^+$) pairs' recombination [114].

In this regard, Chatzisymeon et al. explored the photocatalytic treatment of a three-phase OMWW remediation approach using $TiO_2$ in a laboratory-scale photoreactor. By properly optimizing the contact time, they observed the enhancement of COD removal. The product was a non-toxic effluent with 200 mg·$L^{-1}$ COD organic content [98].

In this context, the high surface/volume ratio of $TiO_2$ nanoparticles, the possibility to dope them to increase the activation under solar irradiation, and the resistance to photo-corrosion are advantages related to the use of $TiO_2$-based photocatalysts.

This hitch can be minimized with the introduction of a second metal oxide component (e.g., $MnO_2$, $NiO$, $La_2O_3$, $SiO_2$, $SnO_2$, $ZnO$, $ZrO_2$), which has been recognized to induce significant degradation under UV irradiation [115–119], generating oxygen vacancies by the substitution of di- or tri-valent atoms by tetravalent atoms and providing particle-particle interaction [120]. In this context, very promising results have been obtained in terms of improved chemical stability and photocatalytic activities of the obtained materials, as demonstrated by many researchers in the last decades [121–123] and recently by Yaacob et al. for $ZrO_2$-$TiO_2$ materials [124].

However, $TiO_2$ has been recently recognized as a carcinogenic substance [125], so an unavoidable challenge is the development of alternative systems able to maintain the same or better photocatalytic activity. In this scenario, among all the potential candidates, one could be zinc oxide (ZnO), which is able to absorb a wide fraction of the solar spectrum and more than $TiO_2$ [126]. Many researchers have demonstrated its efficiency in the photodegradation of organic pollutants in water matrixes [127]. Additional features describe ZnO more than $TiO_2$ [128]; by way of example, it can be used in acidic or alkaline environments through proper treatment [129,130]. Moreover, the optimum pH for the ZnO process is *ca.* 7, whereas that of $TiO_2$ lies at acidic values, implying lower operational costs and higher efficiency than $TiO_2$ in the advanced oxidation of pulp mill bleaching wastewater [131], phenol and 2-phenyl phenol photooxidations [132,133]. In addition, it is highly photosensitive, stable, and possesses a bandgap of *ca.* 3.2 eV [134]. However, besides the numerous studies on using this material in this field, efforts to overcome drawbacks are necessary.

Visible/Solar Photocatalysis

As discussed so far, each step of the industrial sector for olive oil production implies high operational costs. In this context, any improvements introduced to reduce treatment costs must be carefully considered. Among these, for photocatalytic remediation, solar energy has to be properly developed, especially in the Mediterranean countries, with the final aim of cost-effectiveness.

Visible/solar photocatalytic strategies employ adequately designed heterogeneous and homogeneous photocatalysis, photo-Fenton, and solar-Fenton reagents. Some examples are reported in Table 3.

Gernjak et al. investigated OMWW from Portugal and Spain by solar photocatalysis [105]. In more detail, two solar reactors i were employed at pilot scale: (i) a conventional compound parabolic collector type (CPC); (ii) an open non-concentrating falling film reactor (FFR). Different solar photocatalytic systems were tested, but the photocatalyst with the higher amount of Fe (10 mM) showed the most increased activity.

**Table 3.** State of the art in visible/solar photocatalytic processes for OMWW treatment. Adapted with permission from Reference [135].

| OMWW Origin | Type of Process Treatment and Scale | Obtained Results | Ref. |
|---|---|---|---|
| Spain and Portugal | (i) Solar photocatalysis with $TiO_2$ or added peroxydisulphate, or (ii) solar photo-Fenton—pilot plant | (i) Solar photocatalytic systems did not present sufficientefficacy (ii) 85% COD and up to 100% phenols concentration removal | [105] |
| Italy | (i) Centrifugation + solar photolysis, or (ii) centrifugation + solar modified photo Fenton—laboratory scale | (ii) COD and phenolics removal efficiencies up to 29.3% and 63.6% | [136] |
| Italy | Fenton preceded by coagulation—laboratory scale | 85% COD removal (2 h) | [137] |
| Portugal | Biological (fungi *Pleurotus sajor caju*) and photo-Fenton oxidation—laboratory scale | COD removal efficiency up to 76% and total phenols up to 92% | [99] |
| Cyprus | Coagulation–flocculation, extraction of phenolics and post-oxidation by photo Fenton—laboratory scale | COD removal about $73 \pm 2.3\%$ and total phenols of $87 \pm 3.1\%$ | [18] |
| Turkey | Sequential adsorption, biological and photo-Fenton treatment—laboratory scale | 99% phenols reduction and 90% total organic content | [138] |
| Spain | $UV/H_2O_2$—laboratory scale | COD removal of 40–48% (30 min) | [139] |

Ruzmanova et al. studied the photocatalytic treatment of a three-phase OMWW photodegradation process using reusable N-doped $TiO_2$ sol-gel compounds, demonstrating the higher activities of doped-catalysts compared to the non-doped ones, reaching a COD removal more elevated than 60% [140]. Additionally, N-doped materials maintain high efficiency when used for several cycles.

In addition, the role of photochemistry in the Fenton-like process is gaining attention thanks to ultraviolet and/or visible light to reduce the catalyst loading, enhancing the catalytic behaviour. In particular, Gernjak et al. investigated OMWW treatment processes by solar-photo Fenton approach on a pilot-plant scale, successfully removing up to 85% COD and 100% phenols [105].

Andreozzi et al. proposed an OMWW treatment based on a three-phase method exploiting (i) centrifugation followed by solar photolysis, (ii) centrifugation and solar photo-Fenton, and (iii) centrifugation coupled with solar photo-Fenton and ozonation. In this context, the ferric catalyst is responsible for COD and phenol removal (up to *ca.* 30% and 64%, respectively) [136].

Rizzo et al. investigated OMWW treatment by photo-Fenton, preceded by coagulation. In this case, the maximum efficiency of organic matter removal was *ca.* 95% in 1 h [137].

Justino et al. studied the combination of fungi *Pleurotus sajor caju* and photo-Fenton oxidation [99]. The treatment by fungi confirmed the reduction of OMWW toxicity towards *Daphnia longispina* and resulted in 72.9% total phenolic compounds removal and 77% COD reduction. When the treatment is preceded by photo-Fenton oxidation, the biological treatment with fungi is more efficient.

Papaphilippou et al. proposed a treatment process for OMWW by coupling coagulation–flocculation and Fenton oxidation. Following the photo-Fenton oxidation, COD and phenol removals were approximately 73% and 87%, respectively [18].

Finally, Aytar et al. reached 99% phenol and 90% total organic content reduction using adsorption, biological (*T. versicolor*), and photo-Fenton treatment in sequence [138].

Considering the depicted scenarios, it emerges that a proper comparison among the performances of the studied technologies to treat OMWW is not a trivial task. Indeed, the numerous variables in play (i.e., OMWW origin, process type and operative conditions, used scale) do not allow identification of a method that guarantees the best results in terms of OMWW removal. Only a rough evaluation in terms of COD removal can be done, but in this case, all the advantages and/or drawbacks of each strategy must be considered. In general, looking at the COD removal values reported in Table 3, interesting results were

obtained when working on a laboratory scale and in pilot plants, suggesting promising avenues that deserve to be investigated.

## 5. From Conventional to Easily Recoverable Magnetic Photocatalysts

As described in the previous sections, many approaches have been investigated for OMWW treatment [124,141–146]. Still, most of them suffer from not trivial and not negligible drawbacks (i.e., expensive maintenance, lateness in the separation time, high retention time).

In this regard, technologies based on photocatalysis can be advantageous for their environmental friendliness and high oxidation efficiency [147–149]. To develop even more efficient photocatalytic systems for real applications, research continuously moves the efforts toward exploring different materials.

Conventional nano-or micro-powder photocatalysts are developed for continuous, safe, and efficient photocatalytic reactions. Still, at the same time, their use is limited by the difficult separation and recovery from the reaction mixture for their sustainable reuse [150,151]. The recovery cost could invalidate the technology from an economic viewpoint [152]. To overcome this issue, the introduction of magnetic features in photocatalytic systems seems to be one of the best solutions, giving the possibility to maintain the catalytic performances of samples while making their separation from the reaction a more accessible medium.

Several approaches have been recently explored to develop advanced magnetic photocatalytic materials for wastewater remediation. However, unfortunately, few studies have mainly focused on applying these materials in the treatment of OMWW.

For this purpose, different magnetic nanoparticles (i.e., $\gamma$-$Fe_2O_3$, $Fe_3O_4$, $MFe_2O_4$, where M = Mg, Ni, Zn, Cu, Co) have been introduced in photocatalysts, giving rise to composite materials with magnetic features [153–156]. In this context, electron and hole migration between the magnetic and semiconductor components results in the separation of the photo-induced charge carriers, enhancing the light absorption ability [153–156]. This class of innovative materials has been studied regarding several pollutants in wastewater decontamination. Shen et al. prepared $Fe_3O_4@TiO_2@Ag$-Au microspheres with promising magnetic and photocatalytic properties [157]. Singh et al. immobilized $BiOI/Fe_3O_4$ photocatalyst on graphene oxide to degrade 2, 4-dinitrophenol [158]. Furthermore, the potentialities of other magnetic composite photocatalysts have been explored, such as $Cu_2V_2O_7/CoFe_2O_4/g$-$C_3N_4$ [159], $MnFe_2O_4/SnO_2$ [160], $MoO_3/CoFe_2O_4$ [161]. As already mentioned by Ma et al., the research efforts in this field have resulted in the development of several simple and magnetic photocatalytic materials, such as magnetic bismuth-based photocatalysts [162]. In addition, Ruzmanova et al. developed magnetic core $TiO_2/SiO_2/Fe_3O_4$ nanoparticles to degrade organic compounds in OMWW. 1.5 g·$L^{-1}$ of catalyst dosage optimized the photodegradation process, providing high efficiency and an easy catalyst recovery [140]. Successively, Vaiano et al., using ferromagnetic N-$TiO_2/SiO_2/Fe_3O_4$ nanoparticles, achieved 64% phenol removal and 55% TOC reduction after an irradiation time of 270 min, as well as good stability of the photocatalytic materials after four operation/regeneration cycles [163]. Hesas et al. explored a magnetically separable $Fe_3O_4$ on modernite zeolite to purify OMWW from Kermanshah. They identified the key parameters influencing COD and BOD removal: pH (optimized at the value of 7.8) and turbidity of the treated solution. In addition, in this case, the regenerated $Fe_3O_4/mordenite$ zeolite could be reused for five consecutive cycles [164].

In addition, the research community is currently working hard on novel alternatives.

## 6. Perspectives

Considering the high impact of OMWW treatment on the environment and human health, all the sustainability and circular economy principles should be adequately assessed. In this context, perspectives related to the development of efficient, sustainable alternatives to nano- or micro-sized photocatalysts to treat OMWW (Figure 5) can be mainly divided

into two categories: (i) eco-friendly materials (mainly characterized by magnetic features) already investigated in the treatment of several "model pollutants"; and (ii) other emerging eco-friendly materials (floating devices, membranes).

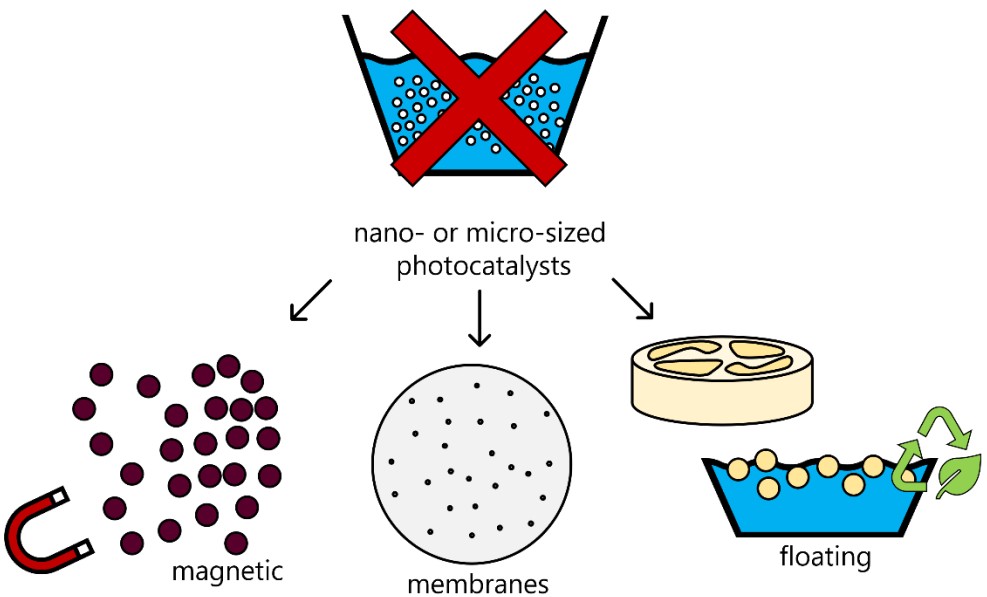

**Figure 5.** Proposed eco-friendly alternatives to nano- or micro-sized photocatalysts to treat OMWW.

*6.1. Eco-Friendly Materials Used to Treat "Model Pollutants"*

Several materials have already been investigated for the degradation of "model pollutants". They show promise for treating OMWW efficiently, and the scientific community could draw inspiration for appropriate evaluations. By way of example, magnetic bismuth-based photocatalysts have been largely used in the wastewater remediation field, and they could also find successful application in OMMW treatment, on which only preliminary studies have been reported.

In general, magnetic bismuth-based materials can be classified as magnetic bismuth-based oxyacid salt, magnetic oxyhalides, magnetic sulfides, and magnetic oxides.

Bismuth-based oxyacid salts (commonly labeled as $Bi_aAO_b$) have gained attention for their excellent visible-light absorption, band potential, and interesting chemical stability [165]. Their specific crystal phase confers good electron transport ability [166]. The introduction of proper magnetic components makes them easily recoverable and reusable for real applications.

In more detail, bismuth ferrite materials ($BiFeO_3$) are characterized by ferroelectricity and ferromagnetic features [167]. They have been explored as nanofibers [168], nanoparticles [169–171], nanosheets [172], nanotubes [173], microspheres [174], and nanorods [175], exploiting their magnetic properties and the 2.2 eV bandgap. Li et al. [173] compared the photocatalytic behaviour of $BiFeO_3$ in the form of nanoparticles, nanofibers, and hollow nanotubes, discovering the superior photoactivity of the latter due to the ultra-thin wall thickness and unique material structure. $BiFeO_3$ nanosheets of 140–230 nm side length and 30 nm thickness were synthesized by Zhu et al. [172] by hydrothermal procedures, demonstrating their high capability to degrade 89% rhodamine B (RhB) under 180 min of visible light irradiation. Bharathkumar et al. [176] prepared $BiFeO_3$ mat and mesh nanostructure materials by an electrospinning method, discovering that the photocatalytic degradation of the mesh sample was greater than that of the mat sample, probably due to the decrease of band gap energy. However, a limitation of the photocatalytic activity of $BiFeO_3$ is related to the fast photogenerated electron-hole recombination. In this context,

some studies pointed out that metal deposition and doping have a positive effect, reducing the charge recombination and improving their resulting photocatalytic performance [177].

Other bismuth-based oxyacid salts with narrow band gaps exist, such as $BiVO_4$ (2.26–2.51 eV), $Bi_2WO_6$ (2.56–2.92 eV), $Bi_2MoO_6$ (2.49–2.66 eV), and $Bi_2O_2CO_3$ (2.8–3.4 eV), which can be combined with magnetic components to obtain interesting and advanced materials with enhanced photocatalytic activity [178,179]. By way of example, Cam et al. introduced $MnFe_2O_4$ on $BiVO_4$, obtaining an innovative material with good photocatalytic activity and magnetic recovery [180]. Sakhare et al. [181] produced $BiVO_4/NiFe_2O_4$ composites able to degrade 98% methylene blue in 240 min of collected sunlight illumination and to maintain excellent stability even after four cycles. Bastami et al. [182] prepared magnetic $Fe_3O_4/Bi_2WO_6$ nanohybrids to degrade ibuprofen under solar light. Xiu et al. [183] developed 3D magnetic $Fe_3O_4/Ag/Bi_2MoO_6$ spheres, obtaining an advanced photocatalytic-Fenton coupling system, which exhibited excellent photocatalytic behaviors in the Aatrex degradation.

Bismuth oxyhalides (BiOX, X = Br, Cl, I) represent another family of bismuth-based materials, which have recently attracted scientific research due to their band gap, high stability, and non-toxicity [184,185]. They exhibit a tetragonal matlockite structure interlaced with $[Bi_2O_2]^{2+}$ flat plates and double halogen atomic layers, which reduce the electron-hole pairs' recombination, producing good photocatalytic behaviour [186,187]. In this context, the combination of BiOX and magnetic components represents an interesting perspective to obtain easily recoverable photocatalytic compounds on which many researchers are working. Briefly, Cao et al. [188] investigated the performances of $BiOBr/Fe_3O_4$ composites, prepared by solvothermal method, under visible light irradiation to degrade glyphosate. Li et al. [189] produced $BiOBr/NiFe_2O_4$ materials of different mass ratios according to a conventional hydrothermal approach, and their photocatalytic performances were explored in the photodegradation of methylene blue and phenol. The authors additionally synthesized BiOBr nanosheets decorated with $NiFe_2O_4$ nanoparticles and tested the samples in the rhodamine-B photodegradation [190], observing that the $BiOBr/NiFe_2O_4$10 (having 10 wt.% $NiFe_2O_4$) composite was able to degrade rhodamine-B more efficiently than the pure BiOBr and $NiFe_2O_4$ (99.8% rhodamine-B degradation after 30 min radiation). Sin et al. [191] prepared N-$BiOBr/NiFe_2O_4$ composites by a hydrothermal strategy, demonstrating the enhanced photocatalytic behaviour towards phenol and Cr(VI) removal.

Moreover, systems based on BiOCl and BiOI were additionally developed, and their photocatalytic performances have been properly investigated. In particular, Ma et al. [192] prepared magnetic $BiOCl/ZnFe_2O_4$ samples, showing their high photocatalytic activity towards penicillin-G degradation (99% penicillin-G degradation within 180 min under visible-light irradiation). Zhou et al. [193] studied ternary magnetic $Ag_2WO_4/BiOI/CoFe_2O_4$ hybrid compounds, evaluating their photocatalytic activity towards toxic elemental mercury Hg(0) removal. In addition, $BiOI/CoFe_2O_4$ composites modified with $AgIO_3$ [194] and $Ag_2CO_3$ [195] were found to be highly efficient in the photocatalytic reduction of Hg(0).

Finally, magnetic sulfides and oxides deserve to be also mentioned. The former (labeled as $Bi_2S_3$) is described by the 1.3 eV energy bandgap and complete visible light region response [196]. They can be combined with materials with magnetic features to promote charge separation and guarantee good recyclability. For example, Li et al. explored the potentialities of $Fe_3O_4/Bi_2S_3/BiOBr$ samples in the photodegradation of diclofenac and ibuprofen, observing *ca.* 94 and 97% conversion of the studied pollutants, respectively, after 40 and 30 min under visible light irradiation [197]. On the other hand, Zhu et al. tested $Fe_3O_4/Bi_2S_3$ microspheres towards Congo red removal, discovering good stability for continuous tests. The latter (commonly named $Bi_2O_3$) is an attractive material possessing high redox reversibility, bandgap spanning from 2.6 to 2.8 eV, and good electrochemical stability [198]. Several researchers combined it with magnetic compounds to obtain final easily recoverable materials. In particular, Abbasi et al. prepared 3D flower-like $Fe_3O_4@Bi_2O_3/g-C_3N_4$ nanocomposites, successively evaluating their photocatalytic activity towards indigo carmine degradation [199]. In this case, introducing the conductive

C layer in the nanocomposite sample could improve the photocatalytic behaviour. In addition, Gao et al. first obtained a $C/Fe_3O_4$ composite and then a double conductive $C/Fe_3O_4/Bi_2O_3$ photocatalyst. In this case, electron-hole pairs' recombination and the reverse electron transfer to $Bi_2O_3$ can be prevented [200].

### 6.2. Other Emerging Eco-Friendly Materials (Floating Devices, Membranes)

Due to their floating properties and good visible light utilization, floating photocatalysts could be considered an excellent choice to gradually substitute conventional photocatalysts [201]. In fact, since 1993, floating $TiO_2$-based materials have been studied [202]. In general, a floating device exploits a lightweight material to float on the water surface, and the photocatalytic performances are maximized thanks to its exposed large surface [203,204]. At the same time, due to its peculiar structure, it minimizes photocatalyst loss, avoiding the long-term contact between photocatalyst and pollutants, which can decrease photocatalytic activity. In the last decades, various supports (i.e., perlite, vermiculite, glass, cork, graphite, polymer) have been investigated as candidates for developing efficient floating photocatalysts [201].

Among them, by way of example, some of the authors studied the performance of aerogel water-floating based materials prepared by poly (vinyl alcohol) and polyvinylidene fluoride as a polymer platform and loaded with different semiconductors, such as g-$C_3N_4$, $MoO_3$, $Bi_2O_3$, $Fe_2O_3$ or $WO_3$, obtaining interesting results towards the reduction of Cr(VI) under visible light [204]. Moreover, Wang et al. [205] recently investigated the use of advanced spongy foam photocatalysts composed of BiOX compounds deposited onto polyurethane foams to degrade targeted pollutants, such as methyl orange, phenol, and chlortetracycline. These systems showed a high potential because they can conjugate high stability, excellent adaptability, and easy recovery, with high photocatalytic performances and good reusability.

In the present panorama, the possibility of using supports characterized by eco-friendly features (i.e., low-cost, non-toxicity, bioavailability) is a priority for further evaluation, and will require strenuous investigation efforts. Some researchers have already considered luffa cylindrica, alginate sphere, or light expanded clay aggregate (LECA), but their potentialities are still the object of study today. Following this perspective, Chawla et al. immobilized $MoSe_2/BiVO_4$ on luffa cylindrica, and then they tested it in phenol degradation, observing up to 97% removal within 2 h of visible light irradiation [206]. Huang et al. recently investigated the possibility of combining alginate spheres with magnetic components, finding exciting results. In this case, the excellent floating performance, together with the availability of reaction sites offered by the material, resulted in the degradation of the selected pollutants (e.g., methyl orange) [207].

Finally, the use of membranes deserves also to be cited. This technology has been investigated in the OMWW treatments for several advantages (simplicity, modulability, easy maintenance, high separation efficiency, small footprint, and easy scale-up) [208]. Several membrane types have been developed and produced, from the polymeric-based ones [209,210] to the inorganic-based ones [211]. All of them have shown excellent performance in the separation of targeted pollutants. However, membrane technology is characterized by some drawbacks. By way of examples, they may be limited by the high concentration of suspended solids present in the OMWW to be treated, and they suffer from foulant deposition due to contaminants separated from the feed. Thus, further treatments are usually required. In this context, Dzinun et al. [212–214] tried to develop a photocatalytic membrane to overcome the membrane fouling and use it as support for photocatalysts. In this case, the photocatalyst addition should minimize the fouling rate. Unfortunately, photocatalytic membranes are also affected by some drawbacks. For example, prolonged exposure to irradiation may ruin their structure, causing damage to the active surface area, which strongly impacts the photocatalytic efficiency [215]. In this context, many ideas are currently put into action by several researchers, as recently reported by Salim et al. [216,217].

All these interesting and promising results obtained in the decontamination of targeted pollutants present in wastewater can be a starting point to investigate more in detail what happens in the case of such complex matrices as OMWW.

## 7. Conclusions

This review provides a critical insight into the current status and the consequent advances related to OMWW treatments, underlying their potentialities and drawbacks. A particular focus on developing innovative eco-friendly photocatalysts, which could become valid alternatives to conventional systems, if properly optimized, is provided.

Nowadays, the OMWW sector plays a fundamental role in the European economy, but at the same time, it also leads to dramatic consequences on the environment and human health. In this context, the current challenge involves optimizing well-known and conventional technologies. Still, the most captivating challenge is the development of innovative advanced strategies, such as those based on photocatalysis. These latter offer many advantages (i.e., high efficiency, low cost) but require the use of novel materials to overcome the common issues related to using slurry reactors and difficult photocatalyst recovery.

In this scenario, the potential use of easily recoverable magnetic compounds as well as floating- and membrane-based devices points to new horizons for sustainability, alternative to conventional $TiO_2$-based systems. The application of these advanced systems still needs hard work by the research world. Their future success in real applications will create a bridge between environmental protection and a circular economy.

**Author Contributions:** Conceptualization, E.F. (Ermelinda Falletta) and C.L.B.; methodology, E.F. (Ermelinda Falletta); visualization, M.G.G., E.F. (Elena Ferrara) and E.F. (Ermelinda Falletta); literature collection and analysis, M.G.G., E.F. (Elena Ferrara) and E.F. (Ermelinda Falletta); Content design, E.F. (Ermelinda Falletta); Writing—original draft preparation, M.G.G., E.F. (Elena Ferrara) and E.F. (Ermelinda Falletta); writing—review and editing, E.F. (Ermelinda Falletta) and C.L.B.; supervision, C.L.B.; project administration, E.F. (Ermelinda Falletta) and C.L.B.; funding acquisition, C.L.B. All authors have read and agreed to the published version of the manuscript.

**Funding:** Velux Stiftung Foundation is gratefully acknowledged for its financial support through project 1381, "SUNFLOAT—Water decontamination by sunlight-driven floating photocatalytic systems".

**Data Availability Statement:** The data that support the plots within this paper are available from the corresponding author on reasonable request.

**Acknowledgments:** This work was supported by the Department of Chemistry, Università degli Studi di Milano, Italy (Piano Sostegno alla Ricerca, PSR, grant 2021).

**Conflicts of Interest:** The authors declare no conflict of interest.

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
