# Peer review of "Olive Mill Wastewater Remediation: From Conventional Approaches to Photocatalytic Processes by Easily Recoverable Materials"

_catalysts, doi:10.3390/catal12080923_

Round 1
Reviewer 1 Report
This review article focuses on the analysis and discussion of an olive oil wastewater treatment process (photocatalysis), after presenting other methods of treating this effluent. The topic is innovative and has a scientific interest, however, some issues should be clarified and/or improved.
1) The purpose of the article and its structure should be clarified in the Introduction section.
2) The authors did not mentionthe olive oil mill wastewater steam reforming as a method of treating this effluent. That method is important and quite recent and should be included in this review.
3) In figure 1, it must be clarified whether the percentage is by mass or volume.
4) There is also a traditional process for producing olive oil. Furthermore, the issue of the 2/3 phases formed in centrifuges is not well explained. When it was used a 2 phase-method, a wet pomace is formed (besides olive oil); when it was considered a 3 phase-method it was generated the pomace (besides the OMWW and olive oil).
5) In Table 1, is it possible to add any parameters of olive oil quality or lower waste production in relation to these technologies? In this table, capitalize or lowercase all.
6) It would be interesting to put tables comparing the performance of various waste treatment technologies. The best results of each method.
7) Confusing sentence: "Non-toxic final effluent with 200 mg·L−1 COD residual organic content and complete TPh removal"
8) TpH ou total of phenols is the same? If yes, use always the same denomination.
9) Is it correct to compare the performance of conjugated processes? Wouldn't it be better to say how much organic load is reduced during the combined process?
10) It was discussed the performance of several materials for the main process of this article. However, some materials are related with treatment processes of others effluents/compositions. It is necessary to clarify the materials already used for the OMWW treatmente or for "model compounds". I suggest to organize the section 5. Besides that, this section is big. I suggest to summarize the information.
11) The authors should discuss in more detail about the negative aspects of the perspectives that are presented at the end of the article, such as concentration gradients in membranes.
12) The conclusions did not focus in the main topic of this article. Please, try to improve this section.
Author Response
We thank the Reviewer for his/her valuable comments.
According to the suggestions, the manuscript was properly modified and the replies to the comments are reported in a file.

Reviewer 2 Report
In this manuscript, a bibliographic overview of the methods for treating olive mill wastewater were reviewed. The development of effective recoverable photocatalysts, which may be a viable alternative to current therapies, is the main emphasis of the review. In indeed, this review provides the useful information for the reader working in this field. The content of the manuscript is relevant and suitable for publishing in Journal of Catalysts. My comments are given as below and hope to improve this manuscript.
1. I recommend that the authors revise the abstract to provide more information on the background, objectives, and conclusion. Please include a phrase demonstrating the importance of the review.
2. The quality of figures should be improved.
3. Introduction part should be improved. The authors should provide more details, such as (a) How does this review vary from others, and (b) What are the goals of this review? (c) What are the present gaps that need to be filled? Since advanced oxidation processes are thought to be efficient ways for treating refractory wastewater, they require further study and explanation. Key references https://doi.org/10.3390/coatings9080465 and https://doi.org/10.1016/j.chemosphere.2019.04.160 .
4. A further read throughout the manuscript to correct minor spelling mistakes is worth doing.
Author Response

(The authors gave the same response as above.)

Round 2
Reviewer 1 Report
Please check the reference 32: it is not well written.
Besides that, the articule it is OK to be published.
Reviewer 2 Report
All the issues raised previously have been satisfactorily answered. I recommend the acceptance of this manuscript in this journal.